# Performance of Sensor Data Process Offloading on 5G-Enabled UAVs

**DOI:** 10.3390/s23020864

**Published:** 2023-01-12

**Authors:** Gerasimos Damigos, Tore Lindgren, Sara Sandberg, George Nikolakopoulos

**Affiliations:** 1Ericsson Research, Laboratoriegränd 11, 977 53 Luleå, Sweden; 2Robotics and AI Team, Department of Computer, Electrical and Space Engineering, 971 87 Luleå, Sweden

**Keywords:** UAV, 5G, offloading, sensors

## Abstract

Recently, unmanned aerial vehicle (UAV)-oriented applications have been growing worldwide. Thus, there is a strong interest in using UAVs for applications requiring wide-area connectivity coverage. Such applications might be power line inspection, road inspection, offshore site monitoring, wind turbine inspections, and others. The utilization of cellular networks, such as the fifth-generation (5G) technology, is often considered to meet the requirement of wide-area connectivity. This study quantifies the performance of 5G-enabled UAVs when sensor data throughput requirements are within the 5G network’s capability and when throughput requirements significantly exceed the capability of the 5G network, respectively. Our experimental results show that in the first case, the 5G network maintains bounded latency, and the application behaves as expected. In the latter case, the overloading of the 5G network results in increased latency, dropped packets, and overall degradation of the application performance. Our findings show that offloading processes requiring moderate sensor data rates work well, while transmitting all the raw data generated by the UAV’s sensors is not possible. This study highlights and experimentally demonstrates the impact of critical parameters that affect real-life 5G-enabled UAVs that utilize the edge-offloading power of a 5G cellular network.

## 1. Introduction

Unmanned aerial vehicles (UAVs) are rapidly developing and they are already being investigated and chosen for a wide range of applications. The unique mobility, site reachability, and emerging ability to interact with the environment have constituted UAVs to be at the center of research for academic institutions and various organizations [1,2]. One central point of the current discussions around UAV applications revolves around use cases that require wide-area connectivity [3,4]. With wide-area connectivity, UAV applications will be less dependent on infrastructure and may even be automated for recurrent assignments. As such, the possibility for automation and infrastructure-free deployments opens a vast field of applications regarding UAVs as a sensing platform. Power line inspection, road inspections, security surveillance, wind power plant inspection, and other potential applications could be some of the most notable ones [5].

Some of the most important benefits that the aforementioned applications will experience revolve around the capabilities of the UAV sensing platform. Cellular-connected UAVs can access the extensive computational power of the edge cloud and can be assisted by centralized processing schemes while maintaining constant connectivity [4,6]. This potential for centralized intelligence makes it possible to handle numerous UAVs or UAV swarms as an interconnected system and perform various types of centralized data processing, constituting the most traditional definition of cloud robotics. The edge cloud promises vast improvements in the real-time execution of computationally intensive algorithms, such as object detection and others [7]. As a result, instead of always being outfitted with powerful computational resources, UAVs can now be developed with fewer computational capabilities and concentrate on equipping such platforms to better serve the purpose of the desired application. The possibilities of cellular-connected UAVs are also expressed by concepts such as extended reality teleoperation and others [8].

For mobile robots, and especially UAVs connected to cellular networks, providing a reliable communication link is a vital prerequisite for a mission to be successful. Challenges and current limitations revolving around this topic have to be thoroughly documented and tested in real-life conditions [6,9]. Theoretical limits and practical integration problems are constantly being examined and evaluated by academia and various organizations involved in this topic. Some of the most notable ones would be the 3rd Generation Partnership Project (3GPP) and European Telecommunications Standards Institute (ETSI). Furthermore, the 5G cellular network offers various improvements to its predecessor, the 4G long-term evolution (LTE). Throughout this study, the capabilities of a 5G-enabled UAV are described, studied, and evaluated by targeting a UAV that is employed as a sensing platform.

Initially, it is best to point out the important factors that affect a 5G-enabled UAV when serving as a sensing platform [5]. One may argue that the round trip latency and jitter measurements are the most crucial key performance indicators (KPIs) for robotics applications, particularly when the transmission of the data packets encapsulates control and command data packets, which thus directly influences the behavior of the robot. Besides that, a crucial characteristic of such applications is the throughput capabilities of the platform. In this article, the uplink and downlink throughput capabilities of the UAV, which is defined to be a user equipment (UE) entity, are analyzed. Overall, 5G offers a significant list of improvements over LTE. Some notable ones would be the 5G core, enhanced uplink and downlink performance, additional quality of service (QoS) features, the ultra-reliable and low-latency communication (URLLC) feature, and so on. This study mainly tries to demonstrate the uplink capabilities of a 5G-enabled UAV and to show the consequences of exceeding the uplink capabilities through practical experiments. From now on, the 5G-enabled UAV sensing platform, which is also a UE in the cellular network, is abbreviated as 5G-UAV.

UAV sensing platforms can be equipped with a large number of sensors. Modern sensors generate a lot of information, and when combined, they can generate amounts of data large enough to surpass the uplink capacity of any modern wireless communication technology. Uplink-heavy sensing platforms, such as robots and UAVs, can typically output data that exceeds the uplink capacity of the 5G network, especially if the vast amount of spectrum in the mmWave range is not utilized. If the 5G-UAV attempts to send more data than the 5G network can handle, the consequences include queue build-up, increased packet latency, and potentially dropped packets. Applications that are regarded as closed-loop applications, such as enhanced teleoperation or autonomous UAVs that rely on external computational resources (e.g., the edge cloud), may exhibit uncertain outcomes. For example, the observed robot state is delayed in the remote endpoint, and such systems should be analyzed as a whole. Therefore, every engineer must carefully assess the capability of the current wireless communication. Then, one has to design a corresponding sensor data handling mechanism to ensure that the application stays within the bounds of the offered communication while making as much use of the available sensor data as possible [7,10].

Here it is important to note that offloading computationally heavy processes from the UAV to the cloud has many benefits. Characteristic examples of such processes are the execution of intensive image analysis algorithms that would require extensive onboard computations that many small-scale UAVs or robots can not comprehend [10]. By offloading computationally heavy processes, less advanced onboard processors can be sufficient, which in turn may reduce the cost of the UAV. In addition, the battery life can be prolonged when less energy is needed for computations. However, offloading relies on the transmission of the sensor data to be processed to the considered cloud. Often, such sensor data transmission requires extensive communication resources and is better suited for human perception rather than computer processing [11]. Similar scenarios are present in different fields, such as smartphone cloud gaming [12] or streaming HD video applications to the cloud for extensive computer vision processing [13,14].

Finally, specifically considering the UAV use cases, it is essential to understand that the sensor data transmission might directly affect the safe operation of a 5G-enabled UAV. For example, the offloading of optimization-based controllers responsible for the positioning of the UAV to the edge cloud, the offloading of perception or path planning algorithms, and the offloading of real-time collision avoidance algorithms to the edge cloud are some crucial examples. Section 3.3 describes such a phenomenon.

The main contributions of this article are summarized as such: (1) A 5G-UAV is evaluated, with a particular emphasis on its uplink capabilities. The utilized architecture employs an edge cloud server to carry out computations for autonomous flights. (2) The 5G-UAV sensing platform and its sensor configuration, while utilizing the robotics operating system (ROS) and its sensor offloading capabilities over 5G, are discussed and evaluated. (3) Improvements in the 5G-UAV communication uplink and downlink interfaces are discussed. (4) Experimental results showing the performance when the sensor data transmitted are well within the limits of the 5G network as well as the consequences of exceeding the limits are provided for the scenarios mentioned above.

## 2. UAV as UE

Many parameters distinguish a UAV from a common smartphone UE in a cellular network. High mobility, not only on the ground level but also at high altitudes, the requirement for higher data rates, acceptable delays on the transmitted packets, and the stringent requirement on the reliability of the transmitted packets are some factors that characterize UAVs as UEs. One can dive deep into subjects such as channel models (free space model or not), interference to/from base stations (BSs), BS handover frequencies, and others, which are important parameters that affect the communication performance of 5G-UAVs [15]. However, the purpose of this study is to demonstrate the capabilities of a 5G-UAV as a sensing platform and to highlight the significant elements and challenges that should be considered when designing and operating applications for 5G-UAVs. The remainder of this section will cover the data flows that occur throughout a UAV’s mission, followed by a description of the capabilities of a 5G-UAV UE.

### 2.1. UAV’s Data Flows

The data flows that appear throughout a UAV mission are critical to the UAV’s successful and safe operation. Therefore, excluding fully autonomous UAVs, data flows impact the mission’s safety and succession must be carefully defined. UAVs must share control-critical information, such as safe landing commands, direction commands, and so forth; this type of information flow is named control and non-payload communication (CNPC) [4,16]. UAVs may also need to communicate mission-related information, such as sensor data, or information that will enable them to collaborate with other robotic agents and result in the mission’s success, this type of information flow is called payload communication (PC) [4,17]. The practical limits for many categories of UAV missions have been thoroughly studied and reported [5]. An illustration of the different UAV data flows is shown in Figure 1. As there are many examples that exhibit the fact that 5G can support the requirements for the CNPC channel of a UAV [5,9,16], the focus of this study is to document the limits of the uplink data flows and the true capabilities of a 5G-UAV sensing platform.

#### UAV Payload Communication

Uplink usage is high in many robotics applications, particularly UAV applications. Many paradigms in the literature and industry, such as collaborative robots, centralized processing, and real-time calculations in the cloud, are constrained by communication delays and uplink throughput. It is vital to understand and analyze how data flows affect mission reliability and the communication boundaries that characterize such applications. As previously described, UAVs that are equipped with many modern sensors produce enough data to exceed the capabilities of any wireless communication system. Engineers must handle sensor data correctly to overcome such constraints and utilize the benefits of wireless cellular communication. Many studies research this topic, such as [10].

During this study, a 5G-UAV is equipped with several sensors and utilized to assess the communication scheme’s capabilities. The aim is to determine which processes can be offloaded to external resources, such as other robotic agents or the edge cloud. The selection of the sensors is inspired by the essential equipment that would enable a fully autonomous UAV. More specifically, the considered paradigm utilizes a 3D LIDAR sensor, an RGB depth color camera, an IMU sensor, and a GPS sensor. Such an autonomy package, together with the possibility of processing sensor data on the edge server and transmitting valuable information back to the platform, would suggest significant improvements in the operation of an autonomous UAV [18]. Section 2.2 presents a theoretical analysis of the 5G-UAV throughput capabilities in the 5G new radio (NR) physical layer. Figure 2 depicts the communication architecture where a 5G-enabled UAV is transmitting sensor data to a remote agent for further processing and storing, in this scenario, the edge cloud constitutes the remote agent.

### 2.2. Cell Uplink Capacity and 5G UE

Cellular networks, particularly 5G networks, differ significantly from other wireless technologies. In particular, 5G is divided into two components the 5G core and the new radio (NR) interface. Here the focus is on the NR interface and its capabilities. Numerous factors contribute to the achievable throughput data rates a UE can achieve when using 5G. The NR interface is a duplex communication system, and the spectrum utilization technique used for the separation of the transmitted signals from received signals is either time division duplexing (TDD) or frequency division duplexing (FDD) [19]. Although FDD was the predominant duplexing scheme in previous-generation cellular networks, TDD configurations now dominate 5G networks. Some of the reasons for the TDD selection are the decreased intra-cell interference and the fact that TDD is considerably more suited for the massive MIMO technology. Nevertheless, some critical parameters affect both TDD and FDD throughput capabilities. The most important ones are the bandwidth of the NR interface, the number of bits per symbol (determined by the modulation scheme, e.g., 256 QAM), and the code rate (R). Thus, Equation (Equation 1) is described in 3GPP TS 38.306 [20] and is used to determine the throughput capabilities of a 5G NR interface.
(1)$$\begin{array}{c} data\;rate\;\left(inMbps\right)=10^{-6}\cdot \sum_{j=1}^{J}(\nu_{Layers}^{\left(j\right)}\cdot \\ Q_{m}^{\left(j\right)}\cdot f^{\left(j\right)}\cdot R_{max}\cdot \frac{N_{PRB}^{\left(BW(j),\mu \right)}\cdot 12}{T_{s}^{\mu }}\cdot \left(1-OH^{\left(j\right)}\right)) \end{array}$$
where *J* is the number of aggregated carriers, $R_{max}=948/1024$, $\nu_{Layers}^{\left(j\right)}$ is the maximum number of supported layers, $Q_{m}^{\left(j\right)}$ is the number of bits per symbol in the maximum supported modulation order, $f^{\left(j\right)}$ is the scaling factor, μ is the numerology, $T_{s}^{\mu }$ is the average OFDM symbol duration in a subframe for numerology μ, $N_{PRB}^{\left(BW(j),\mu \right)}$ is the maximum resource block allocation in $BW_{j}$ with numerology μ, and OH is the transmission overhead. Table 1 provides an example of such a calculation. It also describes the test network used in this study and refers to Section 3, where a collection of results relating to 5G-UAVs is presented. It should also be noted that Table 1 documents the case of a TDD configuration for the uplink resources. TDD separates the uplink and downlink signals in time, and here a 4-to-1 downlink-to-uplink ratio is used. Finally, it is also important to point out that the throughput calculation that Equation (Equation 1) refers to the case that a single UE is utilizing the cell. When multiple UEs are connected to a cell the resources are divided accordingly [19]. Nevertheless, 5G feedback signals can be used to make information, such as the cell load, available to the application layer.

#### UL Enhancements

The 5G standard has many configurations and features that enable uplink-heavy traffic, e.g., it supports uplink-heavy TDD patterns, and the theoretical capabilities are sufficient for handling the majority of uplink-heavy applications, which is also evident in Equation (Equation 1). Nevertheless, there are practical limitations when deploying TDD-based 5G networks. Multi-operator scenarios typically require the synchronized operation of adjacent networks, that is, compatible TDD frame structures have to be adopted among operators to avoid BS-BS and UE-UE interference. A common TDD frame in the mid-band frequencies of 3.4–3.8 GHz is defined to support eight timeslots in the downlink (DL) and two timeslots in the uplink (UL). An example of such a configuration is shown in Table 1 and yields 94 Mbps in the uplink. However, state-of-the-art applications across multiple fields, such as virtual reality, concert entertainment, robotics, machine vision, and the combination of some, such as enhanced teleoperation, are documented to require from 100 Mbps to 10 Gbps in extreme scenarios. For that reason, there are multiple efforts to support enhanced uplink configurations in 5G. The interference mitigation in real 5G deployments will remain the main challenge for realizing TDD uplink-heavy configurations. However, there are multiple solutions that promise significant improvements under the assumption of certain conditions. Uplink-heavy TDD configurations are the most straightforward approach, where the UL frames constitute most of the frame structure. This implementation is also positive in decreasing the latency in the uplink. Yet, that configuration would imply that there would be a wide acceptance for UL-heavy frames since different frame structure configurations will result in cross-link interference. Other solutions capable of increasing UL performance are interference coordination, UL MIMO, carrier aggregation, supplementary uplink, and the use of the unlicensed spectrum NR-U. In general, the majority of the aforesaid solutions require either concrete spatial separation between BSs or rely on agreements between multiple operators. Private networks, such as industrial sites, mines, and others, are strong candidates for the realization of UL-heavy 5G configurations. The reader is referred to the full study for additional information [21].

## 3. Experimental Architecture and Results

The above-mentioned analysis and the capabilities of a 5G-UAV were evaluated using a real 5G stand-alone network. That 5G network is part of an innovation network at the premises of Luleå University of Technology. The network is configured with the parameters given in Table 1. The base station utilized was an Ericsson DOT. The robot is a custom-built quadrotor equipped with the following sensors: a d455 Intel Realsense RGB depth camera, a puck Velodyne lidar, a Vectornav industrial sensor, and two sonar sensors. This sensor combination establishes a platform that can support a wide range of applications, from enhanced teleoperation for beyond visual line-of-sight missions to fully autonomous missions. This 5G-UAV platform can produce a mean uplink data rate of ∼2.4 Gbps and is regarded as a moderate platform in terms of data production, adequate for most modern applications. This section presents experiments considering the aforementioned 5G-UAV sensing platform and examines what are the capabilities of this platform. Initially, stationary experiments with all sensors were carried out without the UAV undertaking any flight-mission operations. Next, an autonomous flight-mission operation was performed and evaluated in two scenarios: (1) with the transmission of sensor data from multiple sensors so that they can be used for process offloading; (2) with intensive background network load.

### 3.1. System Architecture

Figure 2 depicts the system architecture. A 5G-UAV is equipped with a D-link 5G router and is connected to the 5G innovation network. This connection makes use of the 5G local breakout core, which is located quite close to the experiment site. The position of the local breakout core satisfies the delay requirements of real-time robotics applications [9]. An edge server that receives sensor data is co-located with the 5G local breakout core. This edge server is also responsible for the processing of the autonomous mission, which occurs during the second phase of the experiments and is described in Section 3.3 and Section 3.5.

### 3.2. Sensor Offloading—Stationary 5G-UAV

During this experiment, the 5G-UAV platform was transmitting sensor data to the edge cloud processing unit while utilizing the 5G connection. During the stationary experiment, available sensor data were used to determine the capabilities of the current connection. Figure 3 illustrates the uplink capabilities of the 5G network and describes how much data reaches the receiving processing unit at the edge cloud. The streaming data are visualized both in the transmitter’s endpoint and the receiver’s endpoint, and they are illustrated as tx and rx, respectively. The vertical dashed lines differentiate between different experiments, and each bar shows the mean data rate for the transmitted and received sensor data. As expected, only a subset of the available sensor data can be transmitted uncompressed and in real-time across the 5G interface. The uplink capabilities of the 5G network define the limit, which is approximately 94 Mbps, as shown in Table 1. It is important to note that this value corresponds to the uplink throughput achieved at very high signal-to-interference-noise ratio (SINR) values. The receiving end of the architecture experienced significant degradation for the streaming sensor data that exceeded the uplink’s capabilities. This is an expected behavior since the produced data significantly surpasses the uplink capacity of this 5G network. However, when the produced data remains within the uplink limits, the data are transmitted without any degradation.

Similar results are present in the observations regarding the sampling rate of the sensors. Figure 4 illustrates the degradation in the sampling frequency of the offloaded sensors. From these two figures, it is evident that it is not possible to transmit all the data generated by the UAV’s sensors. Therefore, it is preferable to adjust the application layer to make sure that the 5G link is not overloaded. In addition, the mobility of the UAV and the resulting channel quality variations will affect the throughput of the 5G link. To combat these variations, large data buffers can be placed in the 5G-UAV application. Large data buffers in the 5G-UAV would be sufficient to avoid dropped packets in low channel quality conditions but at the price of transmitting delayed packets. Sensor data compression is an obvious candidate that would significantly improve the single 5G-UAV case and make similar applications more scalable when considering multiple agent scenarios. Finally, the offloading of a specific process needs to be carefully evaluated in terms of throughput and latency requirements, the cost of computational resources on the UAV, the onboard power consumption, and others. For example, in [10], an object detection example is presented, where an offloading mechanism is developed and concerns the benefit of the offloading procedure while considering the communication constraints. Table 2 showcase some characteristic offloading cases considering the available sensors on this 5G-UAV platform. As mentioned above, this sensor configuration can support most UAV autonomous missions. Additionally, the expected throughput requirements for the considered sensors are also depicted.

Finally, Figure 5 shows the fraction of the uplink capacity, i.e., here the calculated 94 Mbps (see Table 1), needed for each sensor data type and the required overhead. Here the regarded platform is equipped with a lidar sensor, a depth camera sensor, and an IMU sensor, and it has access to the Vicon capture system, from where the complete state of the robot is provided as ground truth. Furthermore, the pie chart depicts the consumed throughput corresponding to the architecture’s overhead, i.e., the overhead regarding the VPN and the remaining available throughput. Although it is evident that the common 5G network deployment can support a capable UAV platform, it is essential to note that the 23% margin refers to optimal channel conditions and again indicates that in conditions where data KPIs such as low bounded latency and a low rate of dropped packets are essential for the safe operation of a mission, offloading mechanisms must incorporate such information.

### 3.3. Autonomous UAV Mission and Sensor Offloading—Data Traffic Overload

For this experiment, a UAV runs an autonomous mission performing a circular trajectory. The controller responsible for generating the velocity commands for the UAV’s motors is running in a remote edge server, and the communication between the UAV and the edge server is established over a 5G network. Figure 6 depicts the experimental architecture setup. For the intuition behind the UAV model and the utilized controller, the reader is directed to the complete study [18]. The goal here is to test how extensive the degradation of the performance of an autonomous mission is when the uplink of a 5G-UAV is flooded with sensor data far above the maximum throughput of the 5G link. More specifically, the most critical KPI that describes such a mission would be the latency experienced on the application’s control and command data packets [5,9]. The experiment is designed as follows: the 5G-UAV operates while offloading all its computational processes to a remote edge server. While it executes a mission of a circular trajectory, the remote edge server requests the raw sensor data. The 5G network supports both the communication of the sensor data and the control and command data. The transmission of the additional sensor data, in that case, the raw camera image, starts and stops by the operator’s command, and the behavior of the 5G-UAV is observed. Figure 7 and Figure 8 depict the measured latency of the control and command packets in uplink and downlink respectively. Before transmission of the image sensor data is started, the uplink latency has an average value of ∼11 msec. When the transmission of the image sensor data starts, the latency of the control packets significantly increases and peaks at the value of ∼365 msec. Similar behavior is observed for the downlink. This means that the controller processes delayed states of the UAV and thus generates commands that correspond to previous states of the UAV. Hence the overall round trip time (RTT) for the packets that correspond to the UAV is significantly larger than the required value. The consequences of this additional latency are also visible in the trajectory of the UAV (see Figure 9).

### 3.4. Congestion Control

The 5G-UAVs require a meticulous design in all the layers of their architecture. Figure 7 depicts how protocols and, for this experiment, the transport control layer (TCP) protocol, can significantly affect the behavior of the described autonomous 5G-UAV. Although many studies support user datagram protocol (UDP) connections for streaming data, autonomous robotics applications are more complex and require further use-case investigation to validate this claim. Figure 7 and Figure 8 depict the latency behavior observed during data overflow (when the image sensor is enabled for transmission), where the behavior can be explained by the TCP congestion mechanism. More specifically, when data congestion occurs, the TCP congestion mechanism takes place, thus reducing the transmission and affecting the latency of the control data packets. Hence, here it is evident that without a congestion feedback mechanism that UDP does not offer by itself, the delay in the data packets would be even more significant. However, the observed behavior and the corresponding performance can be improved by including congestion feedback mechanisms on top of the transport layer protocols; some promising examples are SCream [22] and L4S [23]. Additionally, it is important to note that the experiments were conducted over stable and reliable physical channel conditions, and the data drop rate was very low and thus considered negligible.

### 3.5. Competing Traffic from Multiple 5G-UAVs

All previous experiments described the behavior of a 5G-UAV operating alone in a 5G cell. Nonetheless, in public cellular networks, single users rarely appear. Therefore, as described in Section 2.2, the available resources of the considered cell are divided between users. The design of this experiment aims to investigate the behavior of a 5G-UAV that offloads a time-critical operation over a 5G network when traffic from a competing 5G-UAV floods the cell’s uplink. The goal here is to observe how the relatively small data rate requirement packets that the 5G-UAV, which runs the time-critical autonomous mission, are affected because of the competing traffic. Note here that, as depicted in Figure 5, the state of the 5G-UAV requires only 0.61% of the cell’s uplink capacity, which corresponds to the entire sensor data needed to operate the autonomous mission. At the same time, the competing agent (the competing 5G-UAV) would like to transmit data rates that surpass the cell’s uplink capability. The experiment starts with the 5G-UAV that runs the autonomous mission of a circular trajectory. At the time $t_{s}$, the second agent initializes an offloading operation that requires more than the cell’s total uplink capacity. Figure 10 depicts the RTT latency measurement for the control data packets of the autonomous 5G-UAV and Figure 11 illustrates the trajectory of the autonomous 5G-UAV. In Figure 10, it is visible that after the initialization of the competing traffic, a small constant offset appears in the RTT control data packet measurements of the autonomous 5G-UAV. The slightly increased latency is associated with the competing traffic and increased scheduling delays. The reason that the 5G-UAV maintains a reliable behavior in its trajectory even when the 5G cell is overloaded by the raw image data is that the 5G network uses a fair scheduler. It is also important to note that for this autonomous mission, the required control data transmission is insignificant compared to the traffic that the second agent requests. Finally, Figure 11 depicts the behavior of the autonomous 5G-UAV, where the accomplished trajectory follows the reference trajectory without any significant error.

Ultimately, to conclude, it is evident that to offload a 5G-UAV mission over the cellular network safely, it should be preceded by a meticulous design that describes the communication requirements of the corresponding application. For example, during this experiment, it was observed that the fair 5G scheduling algorithm ensured a fair splitting of the uplink resources that allowed the mission completion without any significant impact on the trajectory or heavily increased latency. However, note that the requirements for the uplink throughput of the UAV that executed the autonomous mission were significantly smaller than for the second UAV (the one that transmitted raw camera data for offloading). Hence, the dynamic throughput as affected by the total cell load and the mobility must be taken into account in the design; otherwise, as depicted in Section 3.3, delays are to be expected in the uplink traffic of the 5G-UAV. We conclude that offloading use cases should consider the dynamic throughput capabilities of the 5G-UAV and include a sufficient safety margin, as indicated in Section 3.2. It is important to note that there are many works available in the literature that deal either with compression mechanisms or offloading policies [11,13,14], but to the best of our knowledge, not enough that include into their reasoning the mobility characteristic of the 5G-UAV and the resulting outcomes.

## 4. Conclusions

This article examined the capabilities and the limitations of a 5G-UAV serving as a sensing platform. Additionally, the behavior of a 5G-UAV was evaluated in three scenarios: (1) the transmitted data for the corresponding process offloading remains well within the limits of the 5G uplink capacity, (2) the 5G-UAV offloaded sensor data exceeded the cell’s uplink capabilities, and (3) the 5G-UAV was operating an autonomous mission while experiencing significant competing traffic in the same cell. In the first case, the measured latency exhibits robust bounded values and the corresponding application can be reliably served. For the second case, it is evident that a combination of raw sensor data that includes lidars and cameras with too high throughput requirements is not possible to transmit. Applications that rely on the full use of such data for offloading purposes are difficult to serve in a similar architecture. Nonetheless, even if it was possible to transmit that amount of data from a single 5G-UAV sensing platform, this would yield significant scaling concerns. For example, an unofficial report from Intel predicts that in the near future, autonomous vehicles, which serve as sensing platforms and share common characteristics with 5G-UAV sensing platforms, would produce approximately 4000 gigabytes per day [24]. However, it is observed that the combination of the essential sensors often required to manifest an autonomous mission is possible. Modern data compression techniques and intelligent offloading requests enable the possibility of offloading most of the 5G-UAV’s senses to be externally processed. This observation is essential for future innovation, where cellular networks can make applications such as power line inspection, wide-area surveillance, wide-area search and rescue, and others possible while utilizing the existing infrastructure.

Finally, it is essential to note that the operation of a 5G-UAV should be studied and designed as a whole. Many important factors distinguish 5G-UAVs from other traditional UEs. For example, high mobility is an important fact that describes a 5G-UAV, but it could also affect its communication capabilities when served by the cellular network as channel quality varies when the UAV moves between base stations. Additionally, cellular networks such as 5G can provide feedback, helping 5G-UAVs maintain a particular application performance. One characteristic example would be using 5G QoS (data flow prioritization) to prioritize CNPC data over payload data, thus ensuring the reliable transmission of CNPC data.

## Figures and Tables

**Figure 1 sensors-23-00864-f001:**
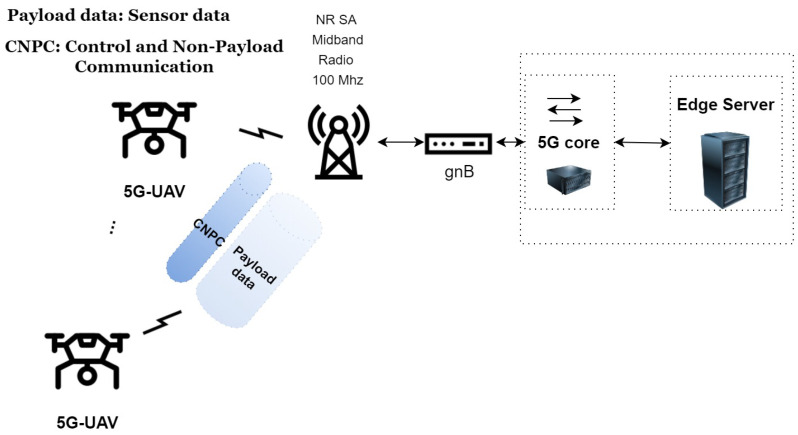
A 5G-UAV, control and non-payload communication, and payload communication.

**Figure 2 sensors-23-00864-f002:**
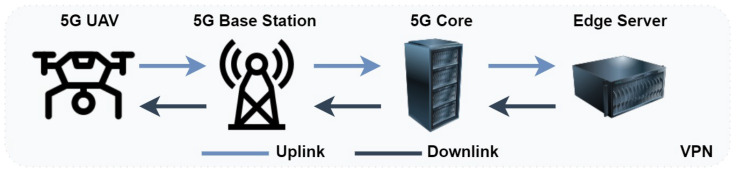
The 5G-UAV (UE) communication system architecture.

**Figure 3 sensors-23-00864-f003:**
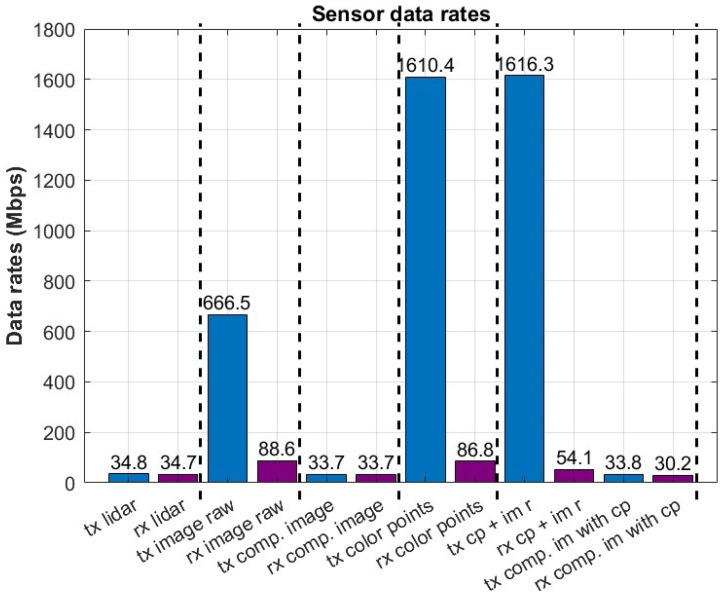
UAV streaming sensors over the 5G cellular network.

**Figure 4 sensors-23-00864-f004:**
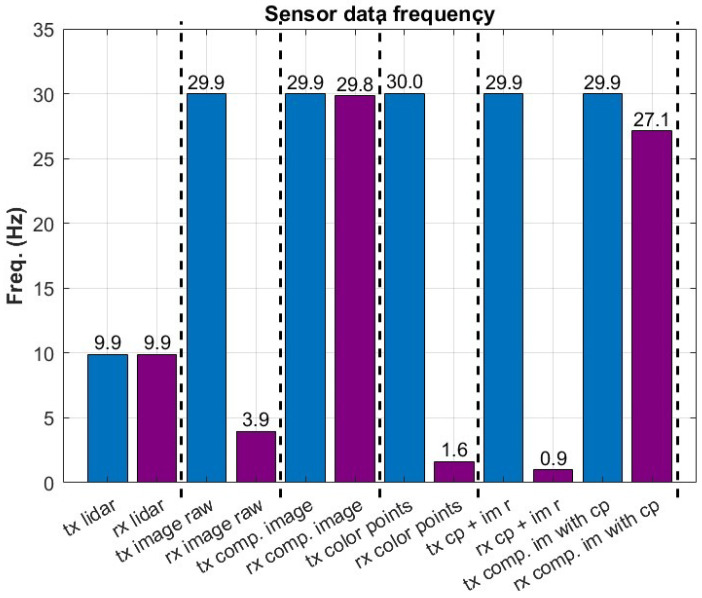
UAV frequency of offloaded sensors.

**Figure 5 sensors-23-00864-f005:**
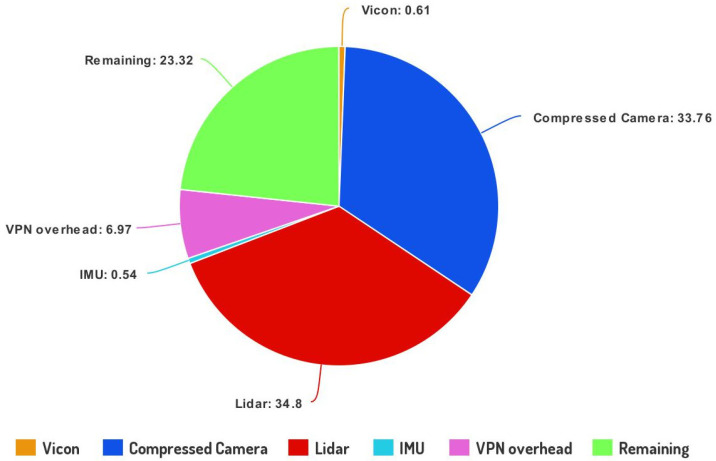
Pie chart that describes the allocation for the average sensor data production of a capable 5G-UAV platform considering the available uplink capacity. Note that the achievable throughput requires high signal strength values and low interference.

**Figure 6 sensors-23-00864-f006:**
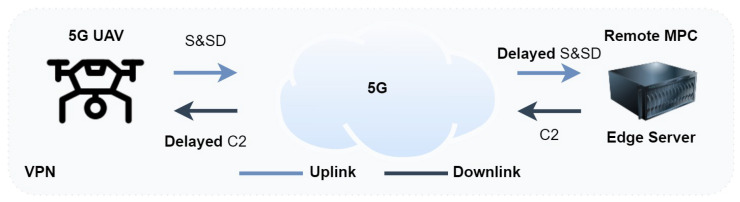
The 5G-UAV autonomous mission architecture. The remote controller is located in the edge cloud. An MPC controller is operating in the cloud and sends real-time control back to the UAV. The communication is established over 5G. S&SD: state and sensor data; C2: control and command data.

**Figure 7 sensors-23-00864-f007:**
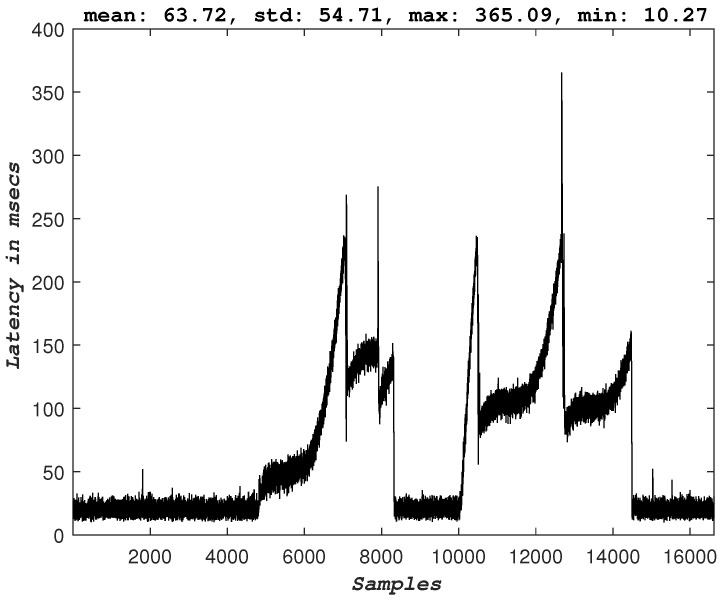
The uplink latency of the 5G-UAV. Sensor offloading on the same 5G-UAV platform is turned on and off two times.

**Figure 8 sensors-23-00864-f008:**
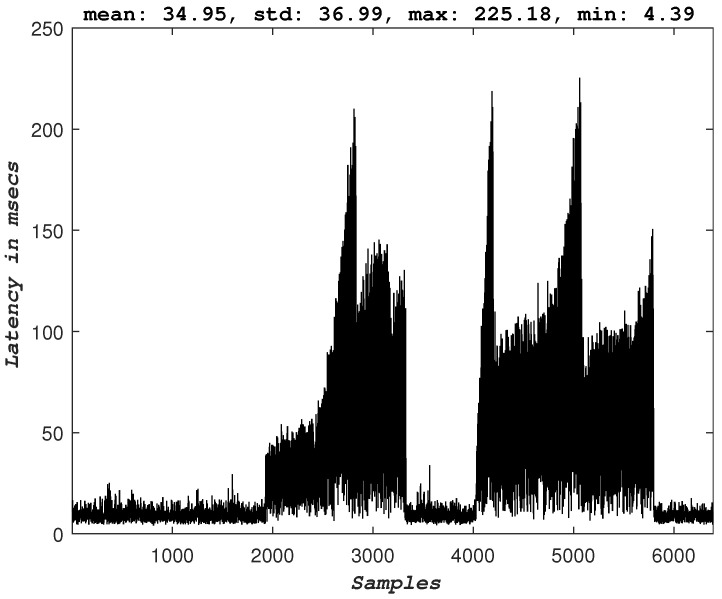
The downlink latency of the 5G-UAV. Sensor offloading on the same 5G-UAV platform is turned on and off two times.

**Figure 9 sensors-23-00864-f009:**
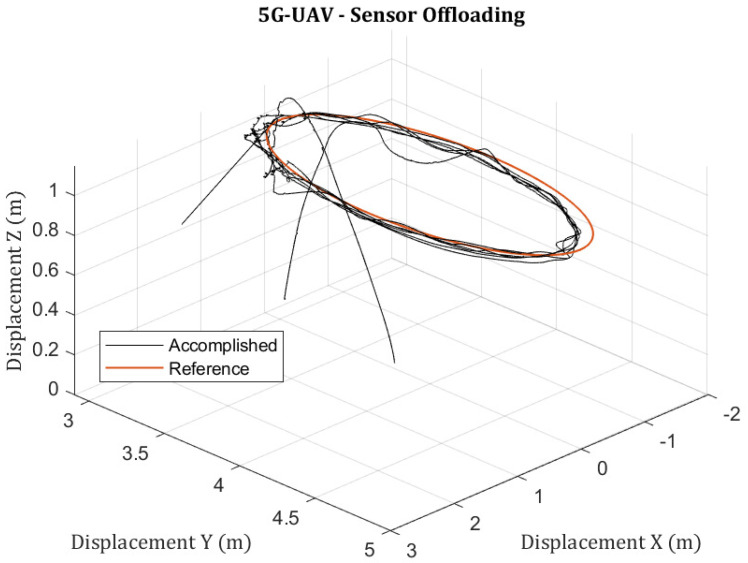
Trajectory of the 5G-UAV. Sensor offloading on the same 5G-UAV platform is turned on and off two times.

**Figure 10 sensors-23-00864-f010:**
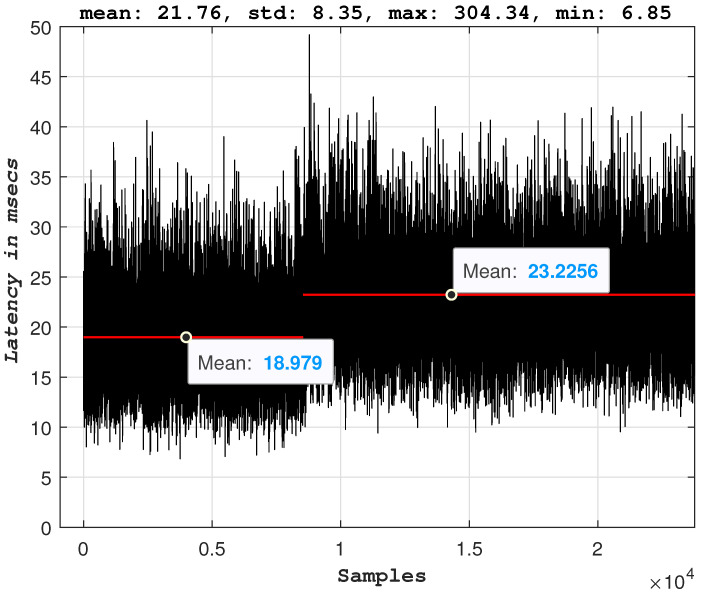
A 5G-UAV’s control data packets RTT when traffic from a competing 5G-UAV begins. A constant offset appears in the RTT latency.

**Figure 11 sensors-23-00864-f011:**
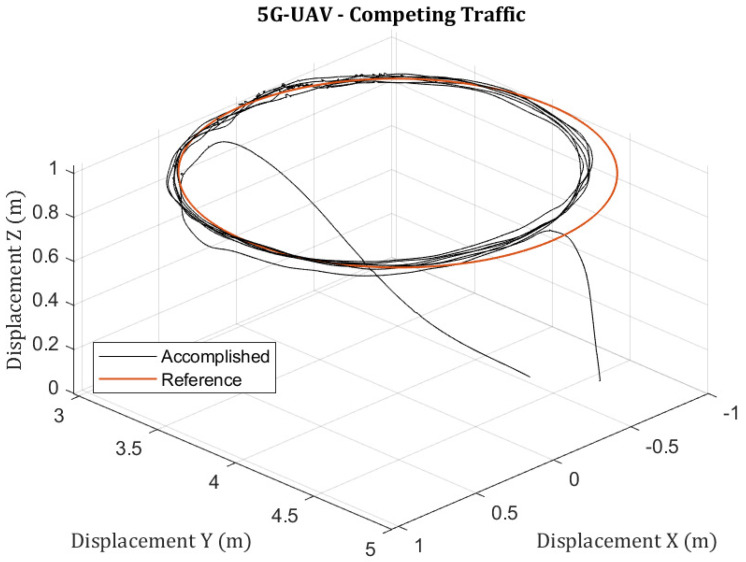
A 5G-UAV’s trajectory when offloading sensors from competing agents or competing 5G-UAVs.

**Table 1 sensors-23-00864-t001:** Utilized 5G innovation network—cell throughput capacity. The values are selected according to the network’s and the UE’s hardware and configuration.

Parameters	Value
Mode	TDD
*J*	1
ν	1 in UL, 4 in DL
*Q*	6 (64 QAM)
*f*	1
BW	100 MHz
OH	0.08 (3GPP 38.306)
DL/UL Ratio	4:1
FR1/FR2	FR1
Downlink	1402 Mbps
Uplink	94 Mbps

**Table 2 sensors-23-00864-t002:** This table shows the average data rates measured from the ROS bandwidth tool in the application layer for various types of sensor data topics. The rightmost column depicts the potential offloading task.

Sensor	Type of Sensor Data	Data Rate	Frequency	Offloading Task
Real Sense	Raw Image	∼667 Mbps	∼30 Hz	object detection, perception algorithms, collision avoidance, optimization algorithms, …
Real Sense	Compressed Image	∼30 Mbps	∼30 Hz	object detection, perception algorithms, collision avoidance, optimization algorithms, …
Real Sense	Depth Point Cloud (RGB)	∼1600 Mbps	∼30 Hz	object detection, perception algorithms, mapping, collision avoidance, optimization algorithms, …
Velodyne	Raw Point Cloud	∼46.5 Mbps	∼10 Hz	mapping, collision avoidance, enhanced teleoperation, …
Real Sense + IMU	State Vector	∼20 Kbps	∼30 Hz	Remote controllers, centralized optimization problems, path planning, localization, …
Velodyne + IMU	State Vector	∼20 Kbps	∼10 Hz	Remote controllers, centralized optimization problems, path planning, localization, …

## Data Availability

Not applicable.

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
