# Peer review of "Performance of Sensor Data Process Offloading on 5G-Enabled UAVs"

_sensors, 2023, doi:10.3390/s23020864_

Round 1

Reviewer 1 Report

The authors proposed Performance of sensor data process offloading on 5G enabled UAVs based approach. The article is well written and adds some significant contribution in the domain. I have following observations

1. The  abstract should be more precise and meaningful. The outcomes and novelty of the method should be reflected here

2. Literature review section should be seperate from introduction. add some more recent works such as

"Improving automated latent fingerprint detection and segmentation using deep convolutional neural network"

"Detection and classification of brain tumor using hybrid feature extraction technique"

3. Figures quality is poor and need to improve

4. How the aggregated carriers are extimated, what is the value authors used in the proposed work

5. Write the simulation settings such as physical channel conditions etc. that are used in this work

6. Results need to elaborate more and put some discussion, comparison with state of art methods

Author Response

You can find our comprehensive reply that addresses all the reviewer's concerns in the attached word document. 

Reviewer 2 Report

Dear authors:

Thank you for giving me the opportunity to read your manuscript. I read your manuscript with great interest. This paper introduced a 5G-based UAV sensor platform. Specifically, as described in the first paragraph in this article, the purpose of this paper is to demonstrate the capacities of a 5G-UAV as a sensing platform and to highlight the significant elements and challenges that should be considered when designing and operating applications for 5G-UAVs.

The structure of this paper is complete and rigorous, but I personally think the manuscript needs some major or minor changes that could significantly improve it. Below you will find some suggestions for what could be done to improve it:

1.      The content of section 3 can be adjusted. Maybe you can consider to change “Case Study - Sensor Offloading” to another better title, because in the following, you first give a short introduction to of the system architecture. Second, I personally think that you can revised the title of section 3.3 to highlight the differences between “5G-UAV static sensor offloading” and “UAV mission and sensor offloading”.

2.      I recommend that the authors can optimize the quality of the figures in the current manuscript because the pictures inside will become blurred when zoomed in. For the tables in this manuscript, I suggest the authors can give separate titles to them.

3.      The author needs to add some additional content to explain the challenges that should be considered when designing and operating applications for 5G-UAVs as this part is poor in the current manuscript.

In conclusion, I personally think that minor revision is needed.

Author Response

(The authors gave the same response as above.)
